# Ecological Risks of Antibiotics in Urban Wetlands on the Qinghai-Tibet Plateau, China

**DOI:** 10.3390/ijerph20031735

**Published:** 2023-01-18

**Authors:** Fengjiao Zhang, Xufeng Mao, Xiuhua Song, Hongyan Yu, Jinlu Yan, Dongsheng Kong, Yinlong Liu, Naixin Yao, Shilin Yang, Shunbang Xie, Haichuan Ji, Huakun Zhou

**Affiliations:** 1MOE Key Laboratory of Tibetan Plateau Land Surface Processes and Ecological Conservation, Qinghai Normal University, Xining 810008, China; 2Key Laboratory of Qinghai Province Physical Geography and Environmental Process, School of Geographical Science, Qinghai Normal University, Xining 810008, China; 3Management and Service Center for Huangshui National Wetland Park, Xining 810016, China; 4Management and Service Center of Qilian Mountain National Park, Xining 810008, China; 5Qinghai Forestry Engineering Consulting Co., Ltd., Xining 810008, China; 6Qinghai Forestry Engineering Supervision Co., Ltd., Xining 810008, China; 7Qinghai Wetland Protection Center, Xining 810008, China; 8Key Laboratory of Cold Regions and Restoration Ecology, Xining 810008, China

**Keywords:** antibiotic risks, source contribution, plateau urban wetlands

## Abstract

Although the ecological risks of antibiotics have been extensively researched globally, fewer studies have been conducted in sensitive and fragile plateau wetland ecosystems. To evaluate the ecological risk of antibiotics in plateau urban wetlands, 18 water samples, 10 plant samples, and 8 sediment samples were collected in March 2022 in the Xining urban wetlands on the Qinghai-Tibet Plateau. The liquid chromatography-electrospray ionization tandem mass spectrometry method was utilized to measure the concentrations of 15 antibiotics in three categories in three types of environmental media. Risk quotients were adopted to assess the ecological risk of antibiotics, and the principal component analysis–multiple linear regression model was used to analyze the source of antibiotics. The results showed that (1) the maximum concentrations of antibiotics in water samples, plants, and sediments reached 1220.86 ng/L, 78.30 ng/g, and 5.64 ng/g, respectively; (2) Tylosin (TYL), norfloxacin (NFX), ofloxacin (OFX), and ciprofloxacin (CFX) in water were at medium and high-risk levels, and OFX had the highest risk value, of 108.04; and (3) the results of source apportionment indicate that 58.94% of the antibiotics came from the Huangshui river and wastewater treatment plant (WWTP) near the wetlands. The current study may provide a reference for the risks and management of antibiotics in plateau urban wetlands.

## 1. Introduction

Antibiotics are widely used in medical, agricultural, and aquaculture industries [1]. There is a growing concern that the overuse and abuse of antibiotics posed an enormous threat to public health and safety [2]. The potential risk of antibiotics has become an important ecological problem [3].

Antibiotics are widespread in the environment and have been detected in surface water, groundwater, plants, soil, sediment, and other environmental media around the world [4,5,6]. Antibiotics are toxic to some organisms and also have a compounding effect on other pollutants [7]. Li et al. showed that microplastics can adsorb antibiotics to produce greater toxicity to aquatic organisms [8]. Likewise, Shi et al. also confirmed that microplastics affected antibiotic concentration [9]. The ecological risks of antibiotics also endanger human survival, with antibiotic resistance causing more death worldwide than the combined impact of acquired immunodeficiency syndrome and malaria [10,11]. Currently, research on the ecological risks of antibiotics in China focuses on the industrially developed and densely populated central and eastern regions, including Taihu Lake [12], Honghu Lake [13], Baiyangdian [14,15], the middle and lower reaches of the Yangtze River [16,17], and the Haihe River [18]. With the increase in human activities in the Qinghai-Tibet Plateau, the impact of antibiotics on plateau ecosystems has received increasing attention [19].

Antibiotic contamination has been studied in different types of wetlands, such as mangroves, lakes, rivers, and coastal wetlands [20,21,22,23,24]. Constructed wetlands have become a hot research area with regard to antibiotics, since constructed wetland systems can improve the removal efficiency of antibiotics and antibiotic resistance genes (ARGs) in the effluent of wastewater treatment plants [25,26]. By optimizing the parameters of constructed wetlands (hydraulic conditions, plant species, etc.), the removal rate can be greater than 98% [27,28,29]. It is worth noting that ARGs can be transmitted through aerosol inhalation and food intake [30,31]. Once mass-resistant organisms enter the environment, they may not be effectively treated because they carry resistance genes, adversely hitting the sensitive and fragile ecosystems [32]. Urban wetlands are critical to the aquatic environment as transitional ecosystems between urban sources of antibiotic pollution and the freshwater environment [33]. Most urban wetlands are built close to living and recreational areas, and although most municipal wastewater is treated by WWTP processes, studies have shown that conventional WWTPs are less effective at removing antibiotics [34]. Thus, levels of antibiotic contamination in urban wetlands remain high.

The Qinghai-Tibet Plateau is a significant gene pool for ARGs [35]. Research on the potential ecological risks of antibiotics in the Qinghai-Tibet Plateau region should be given greater attention, as the ecological environment is more fragile and socioeconomic development is relatively limited. The Huangshui National Wetland Park (HNWP) is located in Xining, the only city on the Qinghai-Tibet Plateau with a population of over a million. A large number of pollutants are discharged into urban wetlands within the park [36]. To further evaluate the potential risk of antibiotics of these urban wetlands, liquid chromatography with tandem mass spectrometry (LC-MS/MS) methods, principal component analysis–multiple linear regression (PCA-MLR) models, and risk-quotient (RQ) ecological risk assessment models were utilized, aiming to reveal the distribution characteristics and sources of antibiotics in various media, as well as the antibiotic risk of the plateau urban wetlands.

## 2. Materials and Methods

### 2.1. Study Area

The HNWP consists of the Huangshui River course and four urban artificial surface flow wetlands, with a total area of 508 hectares. The four constructed wetlands are Haihu wetland, Huoshaogou Wetland, Ninghu wetland, and Beichuan wetland [37] (Figure 1). It is a semi-arid plateau with an annual average precipitation of about 380 mm. About 70% of the rainfall is concentrated between June to September. The average elevation is about 2261 m; low pressure, intense radiation, temperature difference, and aridity are the significant characteristics of the study area [38]. The area around the wetland park is densely populated and commercial, and the source water of wetlands mainly comes from the Huangshui River, Beichuan River, Jiefang Canal, and the reclaimed water from the surrounding wastewater treatment plant (WWTP). *Phragmites australis* and *Typha minima* are the dominant species of aquatic plants [39].

### 2.2. Sampling and Experimental Procedures

A total of 36 samples were collected from 20 sample sites in March 2022, including 18 water samples, 10 aquatic plant samples, and 8 sediment samples (Figure 1, Table 1). All collected samples were transported back to the laboratory and placed in a 4 °C refrigerator.

pH, dissolved oxygen (DO), salinity (SAL) and oxidation-reduction potential (ORP) were determined by HQ40d multiparameter water quality analyzer (HACH, CO, USA). Total nitrogen (TN), total phosphorus (TP), chemical oxygen demand (COD), total organic carbon (TOC), ammonia (NH_3_-N), nitrate (NO_3_-N), orthophosphoric acid (H_3_PO_4_), and heavy metals were detected in the laboratory (GB 3838-2002, GB17378.5 and HJ 832).

Antibiotics were detected by the LC-MS/MS method [40]. The experimental steps are shown in Appendix A. The sources of the antibiotic standards used in the experiment are listed in Appendix A. The mobile phase gradient elution steps are listed in Appendix A. The parameters of the LC-MS assay are listed in Appendix A. The detection limit was 0.01 (ng/L) for water and 0.01 (ng/g) for sediments and plants.

### 2.3. Assessment and Statistical Methods

#### 2.3.1. Enrichment of Antibiotics in Aquatic Organisms

Antibiotics accumulative effects in aquatic plants were calculated using bioconcentration factors (*BCF*) in accordance with Equation (1) [41].
(1)$$BCF=C_{plant}/C_{water}\times 1000,$$
where $C_{plant}$ (ng/g) and $C_{water}$ (ng/L) denote the concentration of antibiotics in plant and water samples, respectively.

Biota-sediment accumulation factors (*BSAFs*) are contaminants in sediments and aquatic organisms and are used to describe the relationship between aquatic organisms and sediment contamination [42]:(2)$$BSAFs=C_{plant}/C_{sediment},$$
where $C_{plant}$ (ng/g) is the concentration of antibiotics in plants and $C_{sediment}$ (ng/g) is the concentration of antibiotics in the sediment.

#### 2.3.2. PCA-MLR Model

A PCA-MLR model was utilized to analyze the relationship between multiple variables through data dimensionality reduction [43,44,45,46]. The equation expressions are as follows:(3)$${\hat{C}}_{sum}=\sum B_{k}FS_{k},$$
(4)$${\hat{C}}_{sum}=\left(C_{sum}-mean\left[C_{sum}\right]/\sigma \right),$$
where ${\hat{C}}_{sum}$ is the standard normalized deviation of the sum of the antibiotic concentrations, *B_k_* is the equation regression coefficient, *FS_k_* is the factor score calculated by the PCA, and σ is the standard deviation of the total concentration of antibiotics.

The formula for mean percentage contribution calculation is $B_{k}/\sum B_{k}$, and the contribution of each source k is derived from Formula (5):(5)$$\mathrm{Contribution}\;\mathrm{of}\;\mathrm{source}\;k\;\left(\mathrm{ng}/L\right)=mean\left[C_{sum}\right]\times \left(B_{k}/\sum B_{k}\right)+B_{k}\sigma FS_{k},$$

Huangshui River (A), Beichuan River (B), Jiefang Canal (C), and WWTP (D) were considered as four sources of pollutants.

#### 2.3.3. RQs of Antibiotics in Surface Water in Wetlands

RQs are used to assess the ecological risk of antibiotic presentation in aquatic environments [47], calculated as Equations (6) and (7):(6)RQ=MEC/PNEC,
(7)$$PNEC=EC_{50}/AF,$$

*PNEC* values were obtained after the literature review (Appendix A) or by dividing the *AF* by the *EC*_50_. The *EC*_50_ is the half of the maximum effect concentration, and *AF* is the assessment factor, with 1000 being acute and 100 being chronic. The ratio of measured environmental concentration (*MEC*) to predicted no-effect concentration (*PNEC*) RQs value was stratified into four levels, ≤0.01, 0.01–0.1, 0.1–1, and >1, indicating insignificant, low, medium, and high risk, respectively.

### 2.4. Statistical Analysis

Microsoft Excel 2016 was used to process the data; R software v4.2.1 (R Foundation for Statistical Computing, Vienna, Austria) and IBM SPSS Statistics version 26.0 (IBM Corp. Armonk, NY, USA) was used for statistical data analysis, and Origin 2021 was used for mapping.

## 3. Results

### 3.1. Antibiotic Concentrations in Different Environmental Mediums

Table 2 shows the concentration of antibiotics in each medium in HNWP. All 15 antibiotics of three categories were detected in water samples. The detection rates of enrofloxacin (EFX), norfloxacin (NFX), lomefloxacin (LFX), difloxacin (DFX), oxytetracycline (OTC), tetracycline (TC), azithromycin (AZM), and tylosin (TYL) were ≥83.33%. The total concentration of antibiotics ranged from not detected (ND) to 1220.86 ng/L. Quinolones (QNs) were the predominant antibiotics with the highest concentrations (0.54–1220.86 ng/L). Ofloxacin (OFX), NFX, and AZM were the main antibiotics in water samples, with mean concentrations of 180.21 ng/L, 56.98 ng/L, and 173.81 ng/L, respectively. The highest and lowest average concentration of antibiotics in water samples were OFX (180.21 ng/L) and RTM (1.62 ng/L), respectively.

Thirteen of the 15 antibiotics were detected in aquatic plants (chlortetracycline (CTC) and doxycycline (DOX) were not detected), and EFX, NFX, LFX, fleroxacin (FLX), DFX, and AZM were detected in 100% of plants. Macrolides (MLs) were the uppermost antibiotics in aquatic plants, and AZM had the highest mean concentration (68.98 ng/L).

Six antibiotics were detected in sediments, and the detection rates of OFX, OTC, CTC, RTM, erythromycin (ERM), and TYL were 100.00%, 12.50%, 100.00%, 87.50%, 100.00%, and 62.5%, respectively. RTM was main antibiotic in sediment, with the highest average concentration of 5.46 ng/L.

### 3.2. Spatial Distribution Changes in Antibiotic Concentration

Figure 2 shows the antibiotic percentage concentrations in different wetlands. MLs was most detected in wetlands, while QNs was most common in water bodies. Although TCs was also detected in the wetland waterbodies, its concentration was the lowest.

In the water samples, the distribution of antibiotics was the highest in the Ninghu wetland (Figure 3a), with the main components being OFX, NFX, and AZM. Moreover, DOX and RTM were detected only in the Ninghu wetland. The most distributed antibiotic in the Beichuan wetland, Huoshaogou wetland, and Huangshui River was AZM. AZM was the most distributed in the Haihu wetland, followed by NFX, OFX, and ERM.

AZM was the most widely distributed in the wetlands (Figure 3b). QNs and MLs were the most distributed in the Haihu wetland, while TCs were distributed only in Haihu and Ninghu wetlands. AZM and TYL were mostly detected in the Huangshui River, the Huoshaogou wetland, and the Beichuan wetland (Figure 3c).

RTM, ERM, and TYL were mostly distributed in the Beichuan wetland and Haihu wetland. TYL was not detected in the Huangshui River and Ninghu wetland, and OTC was detected only in the Huangshui River.

### 3.3. Bioaccumulation and Ecological Risk of Antibiotics

Figure 4a shows the mean BCF values of compounds accumulated by Phragmites australis from the aquatic environment (detailed information is provided in Appendix A). The highest mean BCF values were found for OFX (3691.47 L/kg) and TYL (6659.18 L/kg) in the Haihu wetland; 6659.18 L/kg and 1350.66 L/kg TYL values were found in the Huangshui River and Huoshaogou wetland, respectively; for antibiotics in Beichuan and Ninghu wetlands, the highest mean BCF values were found for NFX (1978.14 L/kg) and LFX (3295.11 L/kg), respectively.

The relationship between Phragmites australis and sediment contamination is presented in Appendix A. The mean antibiotic BSAFs in the three wetlands were >1, except for the RTM. The average BSAFs of antibiotics in Haihu wetland were in the order of TYL (19.49) > ERM (9.17) > OFX (7.2) > RTM (0.78). The average BSAFs of OFX in Huoshaogou wetland was 2.45, and the average BSAFs of antibiotics in Beichuan wetland were in the order of TYL (7.25) > ERM (3.66).

The ecological risk values of antibiotics in wetland water samples are shown in Figure 4b. There was a medium or high risk of antibiotics in the Ninghu wetland and Haihu wetland. FLX, NFX and EFX in the Beichuan wetland were medium risk. The least risk was found in the Huangshui River and Huoshaogou wetland. TYL presented a medium risk in all wetlands, and CFX, NFX, and OFX were at medium and high risk in the Haihu wetland and Ninghu wetland; the risk value of OFX in the Ninghu wetland was as high as 108.04. ERM, AZM, RTM, DOX, OTC, and DFX were low risk, and LFX was very low risk.

## 4. Discussion

### 4.1. Antibiotics Source Analysis of Urban Wetlands

The results of the antibiotics source analysis are shown in Figure 5. Detailed analysis processes could be seen in Appendix A. Four principal components explained 29.6%, 21.91%, 16.025%, and 10.092% variance, respectively, reaching a total of 77.627%. Based on the proportion of antibiotic components, it could be inferred that (1) OFX and NFX had the highest loadings in factor 1, which suitably represent A, C, and D co-sources; (2) DFX, CTC, and RTM had the highest loadings in factor 2, which suitably represent A and D co-sources; (3) AZM had the highest loadings in factor 3, which suitably represents A, B, C, and D co-sources; (4) EFX and CFX had the most loadings in factor 4, which suitably represent the A source.

The linear regression equation of ${\hat{C}}_{sum}$ and FSk for 14 antibiotics is as follows:(8)$${\hat{C}}_{sum}=0.867FS_{1}+0.434FS_{2}+0.17FS_{3}\left(R^{2}=0.962,\;p<0.001\right),$$

Expanding the equation ${\hat{C}}_{sum}$, the equation can be written as follows:(9)$${\hat{C}}_{sum}=0.867FS_{1}\sigma +0.434FS_{2}\sigma +0.17FS_{3}\sigma +mean\left[C_{sum}\right],$$
where σ is 43.596 ng/L and $mean\left[C_{sum}\right]$ is 15.639 ng/L. The average contribution of each factor was estimated to be 58.94% for sources A, C, and D (factor 1); 29.50% for D (factor 2); and 11.56% for sources A, B, C, and D (factor 3).

The highest mean concentrations of OFX (180.21 ng/L) and AZM (173.81 ng/L) were detected in water samples, similar to antibiotic concentrations in urban river systems [38], but dozens of times higher than in natural water bodies [48]. This was in response to the preference of Xining city for antibiotic administration in the winter. Overall, OFX concentrations were higher than those in the lakes in the middle and lower reaches of the Yangtze River (0–106.04 ng/L) [49] and urban rivers in Vietnam (272.0 ng/L) [50]. AZM concentrations were higher than those in the Huangshui River (0.08–7.67 ng/L) [48], associated with their more widespread medicinal use (61.11% and 94.44% detection frequency) and the easy accumulation of slow-flowing antibiotics in wetlands. The minimum value of RTM (ND-1.62 ng/L) was lower than that of wetland Caohai (ND-50.3 ng/L) [51]. In addition, the concentrations of DFX, NFX, and CFX were lower than in Dongguan Reservoir [52].

The cumulative proportion of antibiotics in wetlands was MLs > QNs > TCs; whereas the highest proportion of QNs was found in the water samples from Ninghu, which may be related to the fact that the water source of Ninghu mainly comes from the effluent of the WWTP, while QNs are the most consumed antibiotics in Chinese hospitals [53]. TCs are used mainly in agriculture and livestock; therefore, their lowest concentration was detected in urban wetlands, where domestic wastewater was the main source of pollution [54]. The highest concentrations of MLs were detected in plant and sediment samples, and BCFs and BSAFs calculations also showed that plants had a greater potential for the accumulation of MLs, especially TYL. However, due to the presence of only single species of aquatic plants in urban wetlands and long-term exposure to high concentrations of antibiotic pollution, the damage of the plant membrane system leads to the weakening of accumulation ability [55]; therefore, antibiotic pollution should be prevented and controlled at the source.

Compared to natural wetlands, urban wetlands are more complex sources of pollution, and sediments are soaked for long periods, making a clear distinction between pollution sources difficult. Therefore, the PCA-MLR model was used to analyze the antibiotics sources. The Haihu wetland and Ninghu wetland, with the greatest intensity of human activities, was found to have the greatest source contribution, indicating that human activities had the greatest impact on antibiotics.

The antibiotics were most distributed in the Haihu and Ninghu wetlands, which may be related to the fact that these wetlands are located in the most populated urban areas of Xining. The lower distribution in the Beichuan wetland may be attributed to the improved water quality of the Beichuan River over recent years [37]. Although the Huoshaogou wetland has a high swimming population, its water source comes from the Jiefang Canal, which is used mainly for agricultural irrigation, leaving only a small volume of water [56]. The sampling season coincides with the freezing of water in the Qinghai-Tibet Plateau [57], where the water temperature in the city outskirts is lower than in the city, resulting in insufficient recharge from the freezing and breaking of water flow and a corresponding reduction in the input pollution load. In addition, the OTC in the sediment samples was detected only in the main Huangshui River, which may be because of the extensive pollution sources along the river.

### 4.2. Potential Influence Factors of Antibiotic Concentration

This study uses the Spearman correlation analysis method to explore the relationship between antibiotic concentration and conventional pollutants. Figure 6a shows the correlation between antibiotic concentration and water quality. The correlation coefficients for 11 pairs of conventional water quality parameters were above 0.50, as follows: DO and pH (r = −0.53, *p* < 0.01), SAL and pH (r = −0.52, *p* < 0.01), TN and pH (r = −0.57, *p* < 0.01), NH_3_-N and SAL (r = 0.53, *p* < 0.01), NH_3_-N and TP (r = 0.93, *p* < 0.001), NO_3_-N and pH (r = −0.53, *p* < 0.01), NO_3_-N and TN (r = 0. 83, *p* < 0.001), H_3_PO_4_ and SAL (r = 0.52, *p* < 0.01), H_3_PO_4_ and TP (r = 0.93, *p* < 0.001), H_3_PO_4_ and NH_3_-N (r = 1, *p* < 0.001), COD and ORP (r = 0.61, *p* < 0.01). These results show that the sources of these pollutants are consistent. ΣABs were positively correlated with anthropogenic pollution indicators (ORP, SAL, TN, TP, NO_3_-N, COD, ΣHeavy metals (Hm)) (*p* < 0.01) and negatively correlated with natural property indicators (pH and DO) (*p* < 0.01). This indicates that the source of pollutants in the wetland is acquired and manufactured. Previous studies found that WWTPs and river inputs are the main sources of antibiotic pollution in wetlands [58], and this study also confirms this. Water quality affected the distribution of antibiotics; ΣABs were negatively correlated with DO (r = −0.10, *p* < 0.01), and DO could affect microbial activity and reduce biodegradation [59].

In the sediment (Figure 6b), ERM and CTC (r = 0.80, *p* < 0.01) and TYL and ERM (r = 0.55, *p* < 0.01) were significantly positively correlated, indicating that antibiotics with similar structures are more closely related. ΣABs were negatively correlated with TOC and TN, indicating that these indicators may be affected by more complex factors. ΣHm are significantly and negatively correlated with ΣABs, and although there are few studies on the effect of Hm concentration on antibiotic concentration, most studies have found that Hm pollution promotes antibiotic resistance [60,61]; thus, treating Hm pollution can alleviate antibiotic pollution pressure. The significant positive correlation between ΣABs and pH (r = 0.12, *p* < 0. 01) is consistent with the findings that pH affects the removal efficiency of antibiotics [62].

The significant correlation between antibiotic concentrations in water and sediment and water quality indicators was similar. However, previous studies have shown that antibiotics are transient in water and that persistence in sediment is the dominant factor in antibiotic concentrations [63]. In the treatment of antibiotic contamination, attention should be paid to antibiotic contamination in sediments, which is difficult to remove and can easily form secondary pollution [64]. The ecological risk of antibiotics is closely related to the concentration of antibiotics [50]. The ecological risk of NFX, OFX, and CFX antibiotics was also high in the Ninghu wetland, with high concentration of antibiotics. The ecological risk of OFX reached 108.04. Toxicological data and RQ values of OFX on algae and aquatic animals are shown in Appendix A. The results showed that the toxicity of OFX to rotifers, bacteria, and some algae was high, and the potential risk to crustaceans and some algae (*Pseudokirchneriella subcapitata*) was moderate and low.

## 5. Conclusions

The current study investigated the multiphase occurrence levels of antibiotics in plateau urban wetlands, discussed the spatial distribution and source contribution of antibiotics, and evaluated the ecological risk of antibiotics and the enrichment ability of plants to antibiotics. The main conclusions are as follows:(1)A total of 15 antibiotics were detected in water samples, of which OFX had the highest concentration (0.54–1220.86 ng/L), and 13 and 6 antibiotics were detected in plants and sediments, respectively, with the highest concentrations being that of AZM (68.31–78.30 ng/L) and RTM (5.27–5.64 ng/L).(2)There are significant spatial changes of antibiotics, which are related to wetland water sources and other factors. The highest average concentrations are detected in the Haihu wetland.(3)Compared to other antibiotics, TYL, ERM, and OFX have a higher bioaccumulation potential, and there is a high risk of NFX, OFX, and CFX in water samples.

A limitation of the current study is the lack of seasonal data; the temporal variation of antibiotic ecological risk should be considered in future studies.

## Figures and Tables

**Figure 1 ijerph-20-01735-f001:**
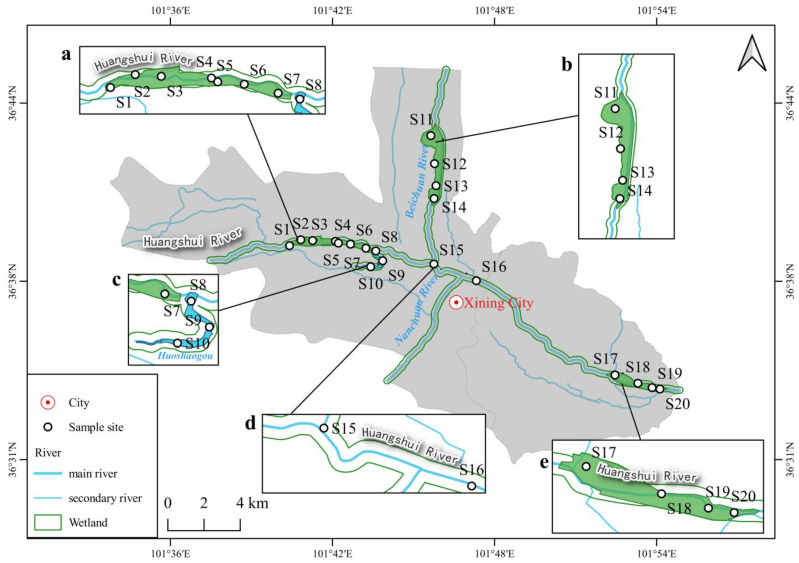
HNWP and sampling sites. (**a**) Haihu wetland, (**b**) Beichuan wetland, (**c**) Huoshaogou wetland, (**d**) Huangshui River, (**e**) Ninghu wetland.

**Figure 2 ijerph-20-01735-f002:**
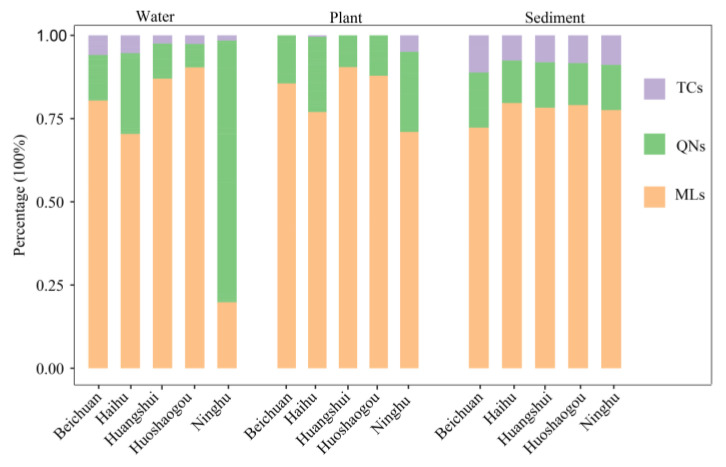
The concentrations of antibiotics in water samples, plant samples, and sediment samples.

**Figure 3 ijerph-20-01735-f003:**
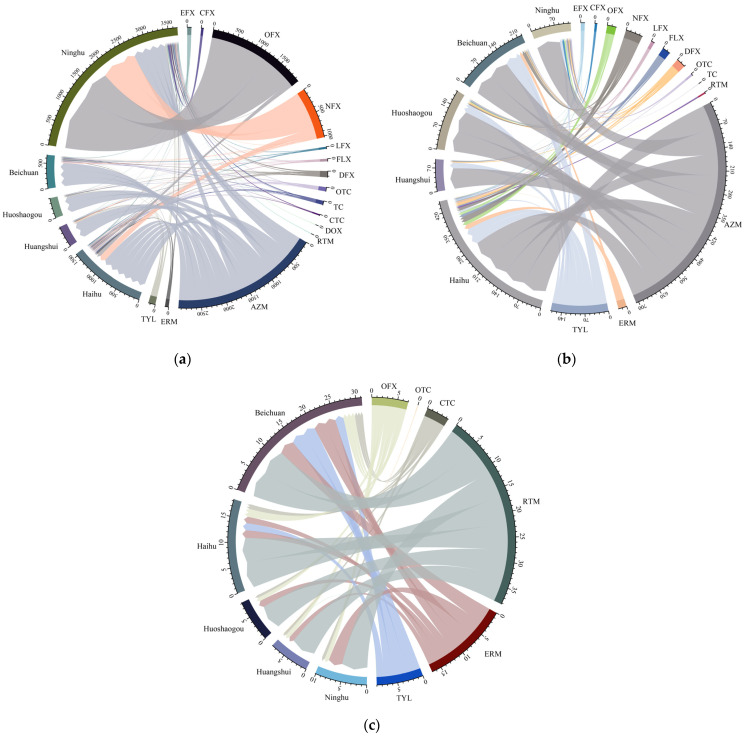
Composition of antibiotics at different wetlands in (**a**) water samples, (**b**) plant samples, and (**c**) sediment samples.

**Figure 4 ijerph-20-01735-f004:**
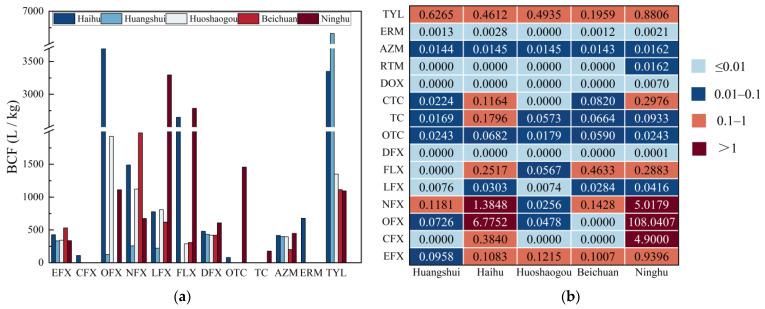
(**a**) Bioconcentration factors (BCF) of antibiotics in HNWP; (**b**) Risk quotients (RQ) of antibiotics in each wetland of HNWP.

**Figure 5 ijerph-20-01735-f005:**
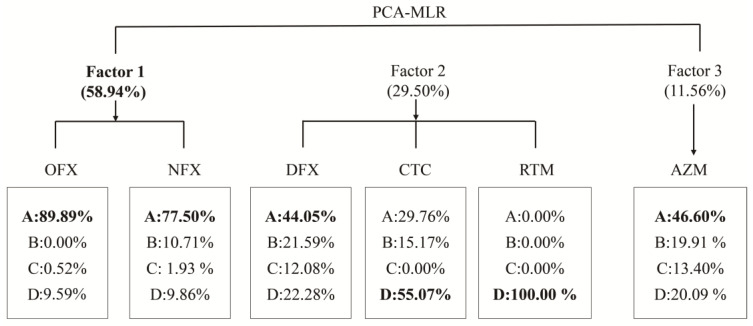
Distribution map of antibiotic contribution rate of HNWP.

**Figure 6 ijerph-20-01735-f006:**
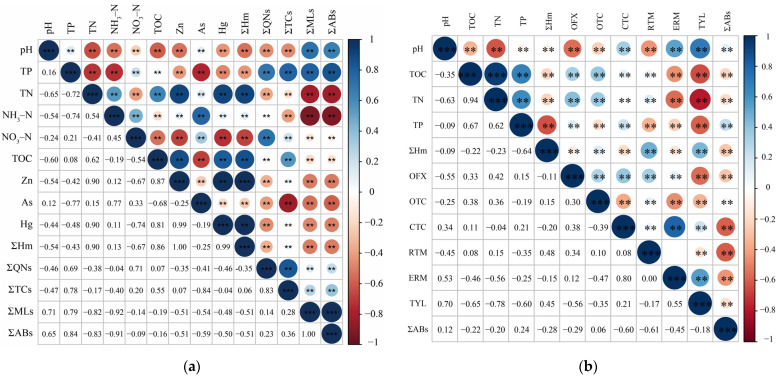
Spearman correlation between antibiotics and (**a**) water or (**b**) sediment properties. Hm included Cr, Cd, Cu, Hg, Ni, Pb, Zn, and As. ** *p* < 0.01, *** *p* < 0.001.

**Table 1 ijerph-20-01735-t001:** Information on latitude, longitude, and type of sampling point.

Sample Sites	Longitude	Latitude	Media	Water Source
S1	101.6735	36.65199	Water, plant, sediment	Huangshui River
S2	101.6805	36.65563	Water, plant	
S3	101.6878	36.65515	Water, plant	
S4	101.7019	36.65466	Water	
S5	101.7036	36.65358	Water, plant	
S6	101.7111	36.65292	Water	
S7	101.7206	36.65041	Water, plant, sediment	
S8	101.7267	36.64874	Water, plant, sediment	Jiefang Canal
S9	101.7310	36.64271	Plant	
S10	101.7236	36.63897	Water	
S11	101.7607	36.71992	Water, plant, sediment	Beichuan River
S12	101.7630	36.70265	Water	
S13	101.7639	36.68901	Water, sediment	
S14	101.7627	36.68109	Water, plant, sediment	
S15	101.7626	36.64064	Sediment	Huangshui River
S16	101.7888	36.63037	Water	
S17	101.8743	36.57224	Water, plant	WWTP
S18	101.8884	36.56714	Water	
S19	101.8972	36.56444	Water	
S20	101.9020	36.56360	Water, sediment	

**Table 2 ijerph-20-01735-t002:** Concentrations of antibiotics in water, plants, and sediments in the HNWP.

Analytes	Water (n = 18, ng/L)					Plants (n = 10, ng/g)					Sediments (n = 8, ng/g)				
	Mean	Fre ^a^	Med ^b^	Min ^c^	Max ^d^	Mean	Fre	Med	Min	Max	Mean	Fre	Med	Min	Max
EFX	4.63	100.00	2.77	2.46	27.06	1.18	100.00	1.23	0.90	1.46	0.00	0.00	ND ^e^	ND	ND
CFX	16.00	16.67	21.58	1.92	24.50	3.53	20.00	3.53	0.84	6.22	0.00	0.00	ND	ND	ND
OFX	180.21	61.11	4.58	0.54	1220.86	4.59	60.00	3.58	0.10	11.50	0.88	100.00	0.87	0.84	0.97
NFX	56.98	100.00	6.29	1.90	520.86	5.15	100.00	4.35	1.27	9.23	0.00	0.00	ND	ND	ND
LFX	2.36	100.00	1.53	0.64	7.74	1.05	100.00	0.77	0.31	2.77	0.00	0.00	ND	ND	ND
FLX	4.07	72.22	3.50	0.84	11.12	2.44	100.00	1.90	0.69	8.47	0.00	0.00	ND	ND	ND
DFX	6.92	100	6.27	5.02	16.14	2.84	100.00	2.81	2.12	3.86	0.00	0.00	ND	ND	ND
OTC	4.91	94.44	3.70	0.68	14.12	2.13	30.00	1.45	0.22	4.72	0.02	12.50	0.02	ND	0.02
TC	5.38	83.33	5.16	0.18	16.16	0.70	10.00	0.70	ND	0.70	0.00	0.00	ND	ND	ND
CTC	5.40	27.78	4.10	1.10	14.88	0.00	0.00	ND	ND	ND	0.56	100.00	0.56	0.51	0.63
DOX	2.22	5.56	2.22	ND	2.22	0.00	0.00	ND	ND	ND	0.00	0.00	ND	ND	ND
RTM	1.62	11.11	1.62	ND	1.62	2.26	20.00	2.26	0.19	4.32	5.46	87.50	5.53	5.27	5.64
AZM	173.81	94.44	172.32	171.34	193.82	70.77	100.00	68.98	68.31	78.30	0.00	0.00	ND	ND	ND
ERM	9.24	44.44	7.63	1.04	17.00	8.48	30.00	9.54	0.74	15.16	2.11	100.00	1.94	1.14	3.41
TYL	8.64	88.89	5.22	2.04	29.94	19.16	90.00	17.81	6.67	39.26	1.80	62.50	1.82	0.02	2.86

^a^ Frequency (%). ^b^ Median. ^c^ Minimum. ^d^ Maximum. ^e^ Not detected.

## Data Availability

Data is contained within the article.

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
