# Peer review of "Ecological Risks of Antibiotics in Urban Wetlands on the Qinghai-Tibet Plateau, China"

_ijerph, 2023, doi:10.3390/ijerph20031735_

Round 1
Reviewer 1 Report
This study evaluated the ecological risk of antibiotics in urban wetlands on the Qinghai-Tibet Plateau, which is important for the study area. However, as the authors said in the manuscript, the ecological risks of antibiotics have been extensively researched globally, except for the uniqueness of the study area, what's new about this study compared with other studies?
1. The resolution of Figure 1 and Figure 6 is too low.
2. Lines 101-106, what is the detection limit?
3. What is the confidence intervals for the principal component analysis-multiple linear regression model?
4. Lines 140-159, are there any differences between the distribution of antibiotic concentration in the study area and that in other wetlands of China?
5. Lines 164-167, why QNs is the most abundant in the water samples from Ninghu?
6. Lines 214-215: even in the same wetland, the sources of pollutants may be different. Is it reasonable that five wetlands were categorized as five sources of pollution? The authors need to explain it.
7. Lines 272-274. Why the authors selected the factors (such as light, pH, TP, TN, Zn, As, Hg, etc.) to analyze their relationships with antibiotic concentrations? The results and analysis in Part 4.2 show little relevance to the topic of the manuscript.
Author Response
Dear Reviewer:
Thank you very much for your work and valuable comments. We have revised the entire manuscript.
Please see the attachment.
Thank you and best regards.

Reviewer 2 Report
The topic and experiments are of high interest to readers. However, certain areas in introduction, materials and methods and discussion requires major changes.

Author Response

(The authors gave the same response as above.)

Round 2
Reviewer 1 Report
- The authors need to check and revise the reference's citation format in the supplementary materials.
Author Response
Dear Reviewer:
Thank you very much for your work and valuable comments. We have revised the reference's citation format in the supplementary materials.
Thank you and best regards.
Reviewer 2 Report
The authors have implemented all the corrections. The manuscript can be accepted for publication.
Author Response
Dear Reviewer:
Thanks very much for your kind work and consideration on corrections of our paper. On behalf of my co-authors, we would like to express our great appreciation to you.
Thank you and best regards.